# Efficacy of Fulvestrant in Women with Hormone-Resistant Metastatic Breast Cancer (mBC): A Canadian Province Experience [note 1]

**DOI:** 10.3390/cancers13164163

**Published:** 2021-08-19

**Authors:** Samitha Andrahennadi, Amer Sami, Kamal Haider, Haji Ibraheem Chalchal, Duc Le, Osama Ahmed, Mita Manna, Ali El-Gayed, Philip Wright, Shahid Ahmed

**Affiliations:** 1College of Medicine, University of Saskatchewan, Saskatoon, SK S7N5E5, Canada; sma936@mail.usask.ca (S.A.); Amer.Sami@saskcancer.ca (A.S.); kamal.haider@saskcancer.ca (K.H.); haji.chalchal@saskcancer.ca (H.I.C.); Duc.le@saskcancer.ca (D.L.); Osama.ahmed@saskcancer.ca (O.A.); mita.manna@saskcancer.ca (M.M.); Ali.elgayed@saskcancer.ca (A.E.-G.); Philip.wright@saskcancer.ca (P.W.); 2Saskatoon Cancer Centre, University of Saskatchewan, 20 Campus Drive, Saskatoon, SK S7N4H4, Canada; 3Allan Blair Cancer Centre, University of Saskatchewan, Regina, SK S4T7T1, Canada; 4Radiation Oncology, Saskatoon Cancer Centre, University of Saskatchewan, 20 Campus Drive, Saskatoon, SK S7N4H4, Canada

**Keywords:** hormone receptor positive breast cancer, fulvestrant, survival, hormone-resistant, visceral metastases

## Abstract

**Simple Summary:**

Fulvestrant is a medication that is approved as first and second-line treatment in patients with hormone receptor positive advanced breast cancer. In clinical practice, fulvestrant is still used beyond the second line of treatment. This study investigated the use of fulvestrant in a Saskatchewan population of women with advanced breast cancer. We found that fulvestrant is effective when used in both the early and later lines of treatment, although the benefit is more pronounced in the earlier line of therapy. Women with disease affecting their visceral organs such as lung, liver or peritoneum had decreased disease control and survival on fulvestrant. Women who had received chemotherapy after fulvestrant and had a clinical response to fulvestrant had better survival.

**Abstract:**

Introduction: Fulvestrant has demonstrated efficacy in hormone receptor positive (HR+) metastatic breast cancer (mBC), both in first-and second-line settings. In clinical practice, however, fulvestrant has been used as a later-line therapy. This study assessed the efficacy of fulvestrant in women with mBC in early-versus later-line therapy. Methods: This retrospective cohort study assessed Saskatchewan women with HR+ mBC who received fulvestrant between 2003–2019. A multivariate Cox proportional survival analysis was performed. Results: One hundred and eighty-six women with a median age of 63.5 years were identified—178 (95.6%) had hormone-resistant mBC, 57.5% had visceral disease, and 43.0% had received chemotherapy before fulvestrant. 102 (54.8%) women received ≤2-line-therapy, and 84 (45.2%) received ≥3 line-therapy before fulvestrant. The median time to progression (TTP) was 12 months in the early-treatment vs. 6 months in the later-treatment group, *p* = 0.015. Overall survival (OS) from the start of fulvestrant was 26 months in the early-treatment group vs. 16 months in the later-treatment group, *p* = 0.067. On multivariate analysis, absence of visceral metastasis, HR: 0.70 (0.50–0.99), was significantly correlated with better TTP, whereas post-fulvestrant chemotherapy, HR: 0.32 (0.23–0.47), clinical benefit from fulvestrant, HR: 0.44 (0.30–0.65), and absence of visceral metastasis, HR: 0.70 (0.50–0.97), were correlated with better OS. Conclusions: Fulvestrant has demonstrated efficacy as both early-and later-line therapy in hormone-resistant mBC. Our results show that women with clinical benefit from fulvestrant, who received post-fulvestrant chemotherapy, or had non-visceral disease, had better survival.

## 1. Introduction

Breast cancer is the most common cancer in women worldwide, with about two million women diagnosed in 2018 [1]. Globally, over 600,000 women died from breast cancer in 2018, with breast cancer being responsible for about 15% of cancer deaths among women [1]. Although breast cancer represents a significant global burden, the mortality rates of breast cancer in high-income countries have declined since the 1980s, largely due to advances in detection and treatment [2].

Breast cancer is divided into three subgroups based on the presence or absence of estrogen receptor (ER), progesterone receptor (PR), or human epidermal growth factor receptor 2 (HER2) overexpression [3,4]. Women with ER/PR positive (HR+) and HER2 negative breast cancer represent over 60% of newly-diagnosed breast cancer [5]. Despite improvements in the outcomes of women with early-stage breast cancer, a subset of women develop metastatic disease. Endocrine therapy is the cornerstone treatment for HR+ advanced breast cancer [6]. Current endocrine therapies include selective ER modulators (SERMs) that act by blocking the estrogen receptor (such as tamoxifen), non-steroidal and steroidal aromatase inhibitors (NSAIs and AIs) that reduce estrogen levels by inhibiting the peripheral synthesis of estrogen (such as anastrozole, letrozole, and exemestane), and the selective ER down-regulator, fulvestrant.

Since endocrine therapy is less toxic than chemotherapy, it is preferable that most women with HR+ breast cancer begin treatment with endocrine therapy. Most women with newly-diagnosed advanced breast cancer are preferentially treated with a non-steroidal aromatase inhibitor (letrozole or anastrozole), with or without a CDK 4/6 inhibitor [7,8,9]. On progression, exemestane (a steroidal aromatase inhibitor), tamoxifen, or fulvestrant are frequently used as potential second-line agents.

Fulvestrant is a pure ER antagonist, exerting selective ER downregulation, and competitively binding to the ER [10,11]. It is administered as an intramuscular injection (500 mg loading dose on days 1, 14, and 29 of the first month, then maintenance dosing monthly at day 28 ± 3 days). The efficacy of first-line fulvestrant in comparison with the aromatase inhibitor, anastrozole, has been demonstrated in the phase III FALCON trial [12]. This trial involved 462 women with metastatic ER-positive breast cancer who had not received previous endocrine therapy. At 25.0 months of follow-up, women who received fulvestrant had a median progression-free survival (PFS) of 16.6 months, versus 13.8 months in those that received anastrozole (HR for progression or death: 0.80, 95% CI 0.64–0.99). Fulvestrant and exemestane are equally active and well-tolerated in women with advanced breast cancer who have experienced progression or recurrence on a non-steroidal aromatase inhibitor [13]. More recently, fulvestrant in combination with targeted therapy in the first and subsequent line settings has shown better outcomes compared to fulvestrant alone [14]. For example, the phase III MONALESSA-3 trial involving 726 postmenopausal women showed that a combination of fulvestrant and ribociclib (a CDK 4/6 inhibitor) in the first- or second-line setting was associated with a median PFS of 20.5 months compared to 12.8 months with fulvestrant alone (HR: 0.59, 95% CI 0.48–0.73). Likewise, the PALOMA-3 trial showed that fulvestrant in combination with palbociclib (another CDK 4/6 inhibitor) in the second or later line therapy was associated with a significantly superior median PFS of 9.5 months compared to 4.6 months with fulvestrant alone (HR: 0.46, 95% CI 0.36–0.59). Fulvestrant in combination with the PI3K3CA inhibitor, alpelisib, in the second or later line treatment has also shown better outcomes with a median PFS of 11.0 months compared to 5.7 months with fulvestrant alone (HR: 0.65, 95% CI 0.50–0.85) [14].

Despite a lack of level 1 evidence of efficacy, in clinical practice, fulvestrant has been used in the 3rd and subsequent lines of therapy in women with ER/PR positive metastatic breast cancer. This retrospective, multicenter cohort study using real-world data explores the efficacy of fulvestrant in women with heavily-treated advanced breast cancer. The primary objective of this study was to compare the efficacy of fulvestrant in women who received fulvestrant as an early line of treatment (≤2 lines of therapy) to the efficacy in women who received fulvestrant as a later line of treatment (≥3 lines of therapy). The secondary objectives were to compare the efficacy of fulvestrant based on: (1) Primary versus secondary endocrine resistance, (2) visceral versus non-visceral metastasis, and (3) to determine prognostic factors that correlate with the benefit of fulvestrant and survival.

## 2. Methods

### 2.1. Eligibility Criteria

This study was approved by the University of Saskatchewan Biomedical Research Ethics Board. Eligible patients were adult women with histologically-documented ER/PR positive breast cancer with metastasis, who had received fulvestrant through a compassionate access program in Saskatchewan provided by AstraZeneca from 2003–2019. Patients were postmenopausal or premenopausal with gonadal suppression. Women who received at least one dose of fulvestrant alone or in combination with a targeted agent were included. In addition, women who had HR+/HER2+ disease and received fulvestrant were included. Individual patient medical records were reviewed, and appropriate data was abstracted with a validated abstraction sheet.

### 2.2. Definitions

Both endocrine therapy and chemotherapy were considered as lines of therapy. Primary endocrine resistance was defined as relapse during the first two years of adjuvant endocrine therapy, or progressive disease within the first six months of first-line endocrine therapy in metastatic breast cancer. Secondary endocrine resistance was defined as relapse on adjuvant endocrine therapy after two years of starting treatment, relapse twelve months after completing adjuvant endocrine therapy, or progressive disease six months after starting endocrine therapy in metastatic breast cancer. Patients with visceral metastasis had evidence of disease in any visceral organs (lung, liver, brain, or peritoneum). Non-visceral metastasis included metastasis confined to the skin and soft tissue, distant lymph nodes, or bones. Clinical benefit was defined as a partial response or stable disease following fulvestrant therapy. Time to progression (TTP) was defined as the time of commencement of fulvestrant until progression of the disease, as defined by the treating oncologist or the last follow-up date. Overall survival (OS) was defined as the time from commencement of fulvestrant until death from any cause or the last follow-up date. Responses were recorded as per the treating physician’s assessment.

### 2.3. Statistical Analysis

Categorical data were summarized as frequency and corresponding proportions. For continuous data, frequency, median, inter-quartile range, mean (as appropriate), and standard deviation were calculated. The chi-square test and Student’s *t*-test were performed for the analysis of categorical and continuous variables. Kaplan–Meier and log-rank methods were used to determine TTP and OS. A multivariate Cox proportional hazard model was used to assess the prognostic significance of various factors that correlate with TTP and OS. For both TTP and OS, the following variables were examined: Previous lines of therapy (≤2 vs. ≥3), age (≤50 vs. >50 years), endocrine resistance (primary vs. secondary), comorbid illnesses, World Health Organization (WHO) performance status (≥2 vs. <2), de novo metastatic breast cancer, chemotherapy use prior to fulvestrant, visceral metastasis, combination therapy (targeted agent) and time from diagnosis of advanced breast cancer to start of fulvestrant treatment. In addition, residence (urban vs. rural), secondary cancer, smoking, clinical benefit (partial response plus stable disease vs. treatment resistant), and use of chemotherapy after fulvestrant were examined for their correlation with OS. The hazard ratio and its 95% CI were calculated. For the variables examined in the final mathematical model, the proportional hazards assumption was assessed using log-log survival curves. The variables that showed *p* < 0.20 on univariate analysis or were considered biologically important were fitted into the multivariate model. The threshold of statistical significance was set at *p* < 0.05. All patients were followed until August 2020, when data entry was closed. SPSS version 27 was used for statistical analysis (IBM, Armonk, NY, USA).

## 3. Results

Two hundred and fourteen women were identified as being registered in the compassionate access program for fulvestrant in Saskatchewan between March 2003 and January 2019. Twenty-eight women were excluded (three women did not receive any dose of fulvestrant, and 25 did not have adequate follow-up information). This study included 186 eligible women with advanced HR+ breast cancer, who had received at least one dose of fulvestrant (Figure 1). The median age at the start of fulvestrant treatment was 63.5 years (IQR: 54.0–74.0). Seventeen (9.1%) women had HR+/HER2+ disease, 81.2% had a WHO performance status of 0–1, and 80.6% of women had a previous diagnosis of early-stage breast cancer, with 89.3% of those having received adjuvant therapy. One hundred and forty-three (76.9%) women had bony metastases, and 57.5% had visceral metastasis. Among the entire cohort, 43% of women had received chemotherapy in the metastatic setting before fulvestrant, and 60.2% of patients received chemotherapy after discontinuing fulvestrant. Patients with HER2+ breast cancer received trastuzumab in combination with chemotherapy or endocrine treatment. Overall, 18.3% of women received fulvestrant in combination with another agent. Of them, 97% received a CDK 4/6 inhibitor. One hundred and seventy-eight women had an endocrine-resistant disease, with 39 (21.9%) having primary resistance and 139 (78.1%) having secondary resistance. Table 1 lists the patient characteristics.

Overall, 102 (54.8%) women had started fulvestrant after ≤2 lines of therapy in the metastatic setting, and 84 (45.2%) women received fulvestrant after ≥3 lines of therapy. Significant differences were noted between the two groups with respect to age, the pathological diagnosis of metastatic breast cancer, median line of therapy, prior chemotherapy, and median time of diagnosis to start of fulvestrant treatment. The median age of women in early-line therapy was 67 years, versus 60 years in women in the later-line therapy group (*p* = 0.001). The median line of therapy in the group that received fulvestrant after ≤2 lines of therapy was 1, compared to 4 in the group who received fulvestrant after ≥3 lines of therapy (Table 2). Women in the ≤2 lines of therapy group started fulvestrant at a median of 24.5 months after diagnosis of advanced breast cancer, compared to 44.0 months in women who received fulvestrant after ≥3 lines of therapy (*p* = 0.002). Women in the early-line group were less likely to have received chemotherapy prior to fulvestrant than the later-line group (17.6% vs. 73.8%, *p* = 0.001). They were also less likely to have fulvestrant-refractory disease (20.6% vs. 38.1%, *p* = 0.009). Overall, 132 (71%) of 186 patients experienced clinical benefit following fulvestrant treatment (Table 2); 81 (79%) of 102 women who received ≤2 lines of therapy compared to 51 (61%) of 84 women who received ≥3 lines of treatment (*p* = 0.006).

### 3.1. Survival

Overall, 74.7% of patients discontinued fulvestrant due to disease progression, 10.2% discontinued due to side effects or patient requests, and 1.6%, due to other reasons. Women in the early-line group were less likely to have fulvestrant-refractory disease (20.6% vs. 38.1%, *p* = 0.009) compared to the later-line group. The median TTP of the entire cohort was 8 months (95% CI: 5.6–10.4). The median TTP was 12 months (9.4–14.6) in women starting fulvestrant after ≤2 lines of therapy, versus 6 months (5.1–6.9) in women with ≥3 lines of therapy (*p* = 0.015) (Figure 2A,B) (Table 3). There was no significant difference in median TTP between women who had primary versus secondary endocrine resistance (7 vs. 9 months, *p* = 0.098). Similarly, women who had visceral metastasis had a median TTP of 7 months compared with 11 months in those with only bone and soft tissue metastatic disease (*p* = 0.142) (Figure 3A,B). Women treated with fulvestrant in combination with a CDK 4/6 inhibitor had a median TTP of 11 months (4.9–17.0) compared to 7 months (5.4–8.6) if they received fulvestrant alone (*p* = 0.11). Women who received chemotherapy before fulvestrant had a median TTP of 6 months (4.5–7.4), versus 12 months (8.7–15.3) in those who did not receive chemotherapy (*p* = 0.039).

The OS of all women following commencement of fulvestrant was 21 months (16.0–26.0). The OS was 26 months (16.0–36.0) in women starting fulvestrant after ≤2 lines of therapy, versus 16 months (10.5–21.5) in women with ≥3 lines of therapy (*p* = 0.067) (Table 3) (Figure 2A,B). There was no significant difference in OS based on primary versus secondary endocrine resistance (*p* = 0.59). Patients that had visceral metastasis had a lower OS, 18 months (14.1–21.9), versus 32 months (23.0–41.0) in those without (*p* = 0.029) (Figure 3A,B). Women treated with fulvestrant in combination with a CDK 4/6 inhibitor had a median OS of 35 months (21.2–48.8) compared to 20 months (15.8–24.2) if they received fulvestrant alone (*p* = 0.07). Patients that received chemotherapy after fulvestrant had a greater OS, 34 months (30.4–37.6), versus 8 months (4.2–11.8) in those that did not (*p* < 0.001) (Table 3).

### 3.2. Cox Proportional Multivariate Analysis

Table 4 lists factors that correlate with TTP and provides the hazard ratios for univariate and multivariate analysis of TTP. On univariate analysis, lack of prior chemotherapy and early-line treatment significantly correlated with better TTP. However, on multivariate analysis, only the absence of visceral metastasis was significantly correlated with better TTP, HR: 0.70 (95% CI: 0.50–0.99). Differences in TTP based on the use of chemotherapy prior to fulvestrant or previous lines of therapy when starting fulvestrant were not significant after adjustment for other variables.

Table 5 lists the hazard ratios for univariate and multivariate analysis of OS, after adjustment of important variables. In multivariate analysis, post-fulvestrant chemotherapy, HR: 0.32 (0.23–0.47), clinical benefit from fulvestrant, HR: 0.44 (0.30–0.65), and absence of visceral metastasis, HR: 0.70 (0.50–0.97), were correlated with better OS. The difference in survival based on WHO performance status was not significant in the final multivariate model after adjustment for other variables.

## 4. Discussion

Our results show that fulvestrant is effective in both early- and later-line therapy in advanced HR+ breast cancer. Likewise, women with both primary and secondary endocrine-resistant disease benefited from fulvestrant. As anticipated, the benefit was more pronounced in women who received fulvestrant as an early-line treatment compared to those who received it as a later-line of treatment. Our real-world study cohort treated with fulvestrant had a TTP of 8 months and an overall survival of 21 months. This is comparable to the outcomes of patients in the CONFIRM trial, in which participants who were treated with second-line 500 mg of fulvestrant had a progression-free survival of 6.5 months and an overall survival of 26.4 months [15,16].

In our study, the group of patients who received fulvestrant as an early-line therapy had a significantly better TTP of 12 months, compared to 6 months in those who received fulvestrant as a later-line therapy; however, on multivariate analysis, this difference was not significant after adjustment of other prognostic variables. Other studies have shown a positive correlation between previous lines of therapy and survival [17,18,19]. For example, a Japanese study showed a 20% relative improvement in time to treatment failure associated with a line of therapy; however, the numerical differences based on treatment line were small, 5.8 months in first- and second-line fulvestrant treatment and 4.6 months beyond the fourth line [17]. Likewise, our results showed that women who received fulvestrant as an earlier line of therapy had a trend of better overall survival than those who received fulvestrant as a later line of therapy (26 vs. 16 months), with a 26% reduction in mortality. This trend towards better survival in women who received fulvestrant as an early-line therapy may be because the patients starting fulvestrant at a later line of therapy were heavily pre-treated in comparison, and had a longer-standing disease that may not respond as well to subsequent therapy. Patients taking fulvestrant after ≤2 lines of therapy had started fulvestrant after a median duration of 24.5 months after being diagnosed with metastatic breast cancer, and had a median overall survival of 26 months after starting fulvestrant. In comparison, patients taking fulvestrant after ≥3 lines of therapy had been diagnosed with metastatic breast cancer for a median duration of 44 months before starting fulvestrant, with a median overall survival of 16 months. Interestingly, patients starting fulvestrant at a later line of therapy had a better overall survival from the time of the diagnosis of advanced breast cancer (median OS of 73 months vs. 48 months). However, this difference in overall survival from the time of diagnosis may be because the group of patients taking fulvestrant as a later line of therapy were younger and had a high rate of chemotherapy prior to commencement of fulvestrant.

Similar to the FALCON trial, fulvestrant was found to be more effective in patients with non-visceral metastases [12]. Our results show that TTP and OS were shorter in patients with visceral metastases compared to those with bone and soft tissue metastases. Other real-world studies have also shown decreased disease control with fulvestrant in patients with visceral metastases [18,19,20,21]. It is important to note that patients with primary versus secondary endocrine-resistant disease did not have significant differences in TTP or OS. Other studies have demonstrated a decreased response to fulvestrant based on previous endocrine insensitivity [18,22]. This study also shows that patients taking chemotherapy after discontinuing fulvestrant had a significantly longer median overall survival of 34 months, compared to 8 months in patients who did not receive chemotherapy, showing a 68% relative reduction in mortality. This observation most likely reflects the importance of chemotherapy in patients with endocrine-resistant cancers and good performance status. In addition, independent of the use of chemotherapy following progression on fulvestrant, clinical benefit from fulvestrant (defined as a partial response or stable disease) was strongly correlated with better overall survival, with an approximately 56% relative reduction in mortality compared to those who progressed on fulvestrant.

Despite the considerable success of fulvestrant, one limitation is that it must be administered by intramuscular injection. Therefore, there is a need to improve the delivery of fulvestrant by developing orally bioavailable selective estrogen receptor degraders (SERDs). Current oral SERDs in phase III development include Elavcestrant (ClinicalTrials.gov Identifier: NCT03778931, accessed 17 August 2021) and GDC-9545 (NCT04546009, accessed 17 August 2021). Although our study participants were primarily using fulvestrant monotherapy, based on promising new data on CDK 4/6 inhibitors, it is recommended that CDK 4/6 inhibitors be added in combination with patients who have not previously received them. The benefit of CDK 4/6 inhibitors to fulvestrant treatment has been well established by significant increases in both PFS and OS [6,14]. In our study cohort, women treated with fulvestrant in combination with a CDK 4/6 inhibitor had better TTP and OS. However, due to a small number of women who received combination therapy and lack of power, the differences did not reach statistical significance.

Our study provides information on the efficacy of fulvestrant in real-world clinical practice; however, it is important to highlight some limitations. First, it was a retrospective study, which carries some limitations. The study population was treated over several years. Only about 20% of patients received combination therapy, so this may not adequately reflect the recent practice of combination treatment, such as the addition of CDK 4/6 inhibitors to the standard of therapy. Another limitation is that the endpoints were based on the treating physician’s determined assessment, which can be difficult to standardize. This study was not able to investigate the effectiveness of fulvestrant in comparison to other treatment arms. However, one of the major strengths of this study is that it had a generally inclusive criteria, and all the women who received fulvestrant and had adequate follow-up in Saskatchewan were assessed, reflecting a population-based study. The results of this study further validate the effectiveness of fulvestrant in previously treated HR+ breast cancer, and provide useful information on the efficacy of fulvestrant based on patient characteristics.

## 5. Conclusions

Fulvestrant has demonstrated efficacy as both an early and later-line therapy in metastatic breast cancer. The OS in both early and later lines of therapy is similar, but women who received ≤2 lines therapy prior to fulvestrant had a better TTP after starting fulvestrant. Women with visceral disease at commencement of fulvestrant, regardless of previous lines of treatment, had a shorter duration of disease control and OS. In addition, clinical benefit from fulvestrant and the use of chemotherapy following fulvestrant were correlated with better OS. This study showed no difference in TTP or OS based on endocrine sensitivity.

## Figures and Tables

**Figure 1 cancers-13-04163-f001:**
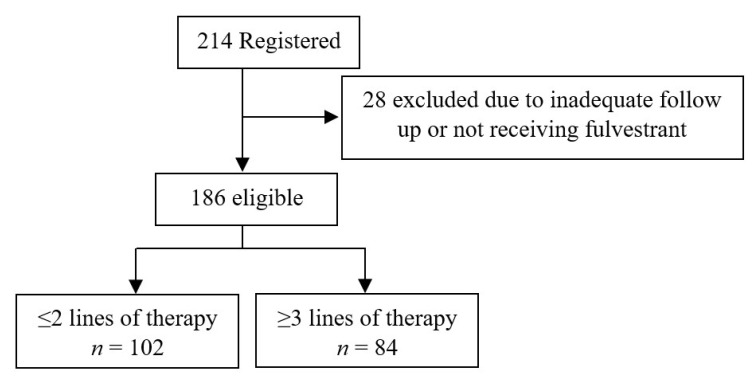
Flowchart of study participants who received fulvestrant.

**Figure 2 cancers-13-04163-f002:**
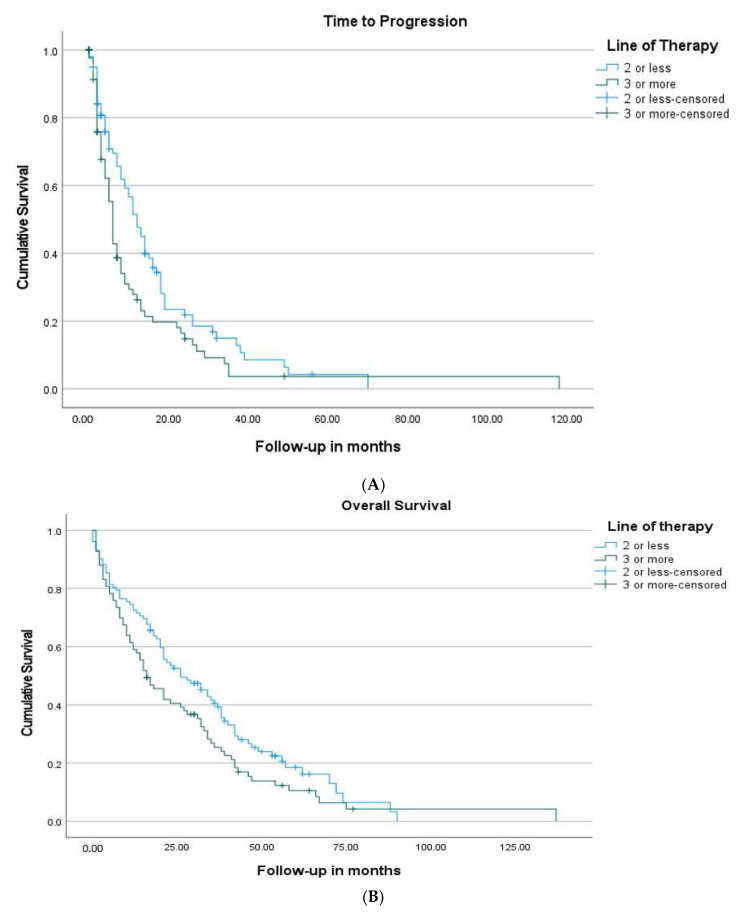
(**A**) Kaplan–Meier survival curves for time to progression stratified by line of therapy with a median TTP of 12 months (9.4–14.6) in women starting fulvestrant after ≤2 lines of therapy, versus 6 months (5.1–6.9) in women with ≥3 lines of therapy (*p* = 0.015). (**B**) Kaplan–Meier survival curves for overall survival (OS) stratified by line of therapy with a median OS of 26 months (16.0–36.0) in women starting fulvestrant after ≤2 lines of therapy, versus 16 months (10.5–21.5) in women with ≥3 lines of therapy (*p* = 0.067).

**Figure 3 cancers-13-04163-f003:**
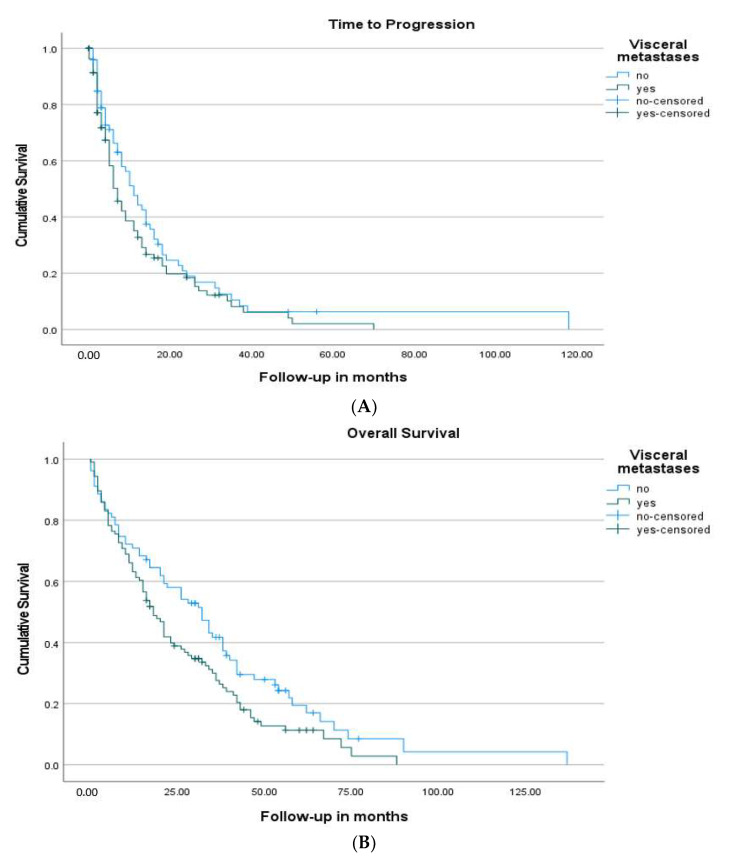
(**A**) Kaplan–Meier survival curves for time to progression (TTP) stratified by the presence of visceral metastasis. Patients who had visceral metastasis had a TTP of 7 months (5.1–8.9), versus 11 months (7.9–14.1) without visceral metastasis (*p* = 0.142). (**B**) Kaplan–Meier survival curves for overall survival (OS) stratified by the presence of visceral metastasis. Patients who had visceral metastasis had a lower OS of 18 months (14.1–21.9), versus 32 months (23.0–41.0) without visceral metastasis (*p* = 0.029).

**Table 1 cancers-13-04163-t001:** Characteristics of patients in the entire cohort and subgroups.

Variables	Study Cohort *n* = 186 (%)	≤2 Lines of Therapy *n* = 102 (54.8%) (%)	≥3 Lines of Therapy *n* = 84 (45.2%) (%)	*p* Value
Demographics				
Median age	63.5 (IQR: 54.0–74.0)	67 (56.7–77.3)	60 (51.0–68.0)	0.001
>50 years	157 (84.4)	93 (91.2)	64 (76.2)	0.007
Rural residence	94 (50.5)	53 (52)	41 (48.8)	0.76
Comorbid illness	94 (50.5)	58 (57)	36 (42.8)	0.24
Secondary cancer	22 (11.8)	14 (13.7)	8 (9.5)	0.49
WHO performance status 0–1	151 (81.2)	82 (80.3)	69 (82.1)	0.85
Smoking History	69 (37.1)	44 (43.1)	25 (29.8)	0.21
History of early-stage breast cancer	150 (80.6)	85 (83.3)	65 (77.4)	0.35
Bilateral breast cancer	12 (8.0)	10 (11.7)	2 (3.0)	0.12
Surgery for early-stage breast cancer				
Mastectomy	94 (62.6)	53 (62.3)	41 (63.0)	0.37
Lumpectomy	36 (24.0)	19 (22.3)	17 (26.1)	0.70
Bilateral mastectomy	20 (13.3)	13 (15.3)	7 (10.7)	0.33
Stage of early-breast cancer				
I	35 (23.3)	18 (21.1)	17 (26.1)	0.55
II	71 (47.3)	40 (47.0)	31 (47.8)	1.0
III	44 (29.3)	27 (31.7)	17 (26.1)	0.47
Received Adjuvant therapy	134 (89.3)	75 (88.2)	59 (90.7)	0.79
Adjuvant chemotherapy	81 (54.0)	44 (51.7)	37 (56.9)	0.62
Adjuvant endocrine therapy	110 (73.3)	67 (78.8)	43 (66.2)	0.09
Adjuvant radiation therapy	96 (64.0)	53 (62.4)	43 (66.2)	0.73
Diagnosis of metastatic disease				
Clinical	39 (21.0)	30 (29.4)	9 (10.7)	0.002
Pathological	147 (79.0)	72 (70.6)	75 (89.3)	0.004
Receptor status				
Estrogen receptor positive	183 (98.4)	100 (98.0)	83 (98.8)	1.0
Progesterone receptor positive	152 (81.7)	84 (82.4)	68 (81.0)	0.85
HER2 overexpression	17 (9.1)	7 (6.9)	10 (12.3)	0.30
Visceral metastases	107 (57.5)	53 (52.0)	54 (64.3)	0.10
Location of Metastases				
Bone	143 (76.9)	75 (73.5)	68 (81.0)	0.29
Lung	78 (40.9)	39 (38.2)	39 (46.4)	0.29
Liver	58 (31.2)	27 (26.5)	31 (36.9)	0.15
Skin or soft tissue	47 (25.3)	22 (21.6)	25 (29.8)	0.23
Nodal	27 (14.5)	10 (9.8)	17 (20.2)	0.06
Brain	7 (3.8)	2 (2.0)	5 (6.0)	0.24
Peritoneal	5 (2.7)	3 (2.9)	2 (2.4)	1.0

**Table 2 cancers-13-04163-t002:** Treatment information and response to fulvestrant.

Variables	Study Cohort *n* = 186 (%)	≤2 Lines of Therapy *n* = 102 (54.8%) (%)	≥3 Lines of Therapy *n* = 84 (45.2%) (%)	*p* Value
Median line of therapy	2 (range 0–8)	1 (0–2)	4 (3–8)	<0.001
Four or more lines of therapy	42 (22.5)	0	56 (71.8)	<0.001
Median time from diagnosis to start of fulvestrant	34 (IQR: 20.0–63.0)	24.5 (12.0–44.5)	44 (28.0–87.0)	0.002
Received chemotherapy prior to fulvestrant	80 (43.0)	18 (17.6)	62 (73.8)	0.001
Combination treatment	34 (18.3)	21 (20.6)	13 (15.5)	0.44
Targeted Therapy	33 (17.7)	21 (100)	12 (92)	0.32
Endocrine Resistance	178 (95.7)	95 (93.1)	83 (98.8)	0.07
Primary	39 (21.9)	27 (28.4)	12 (14.4)	0.04
Reason for discontinuation of fulvestrant				
Progression	139 (74.7)	71 (69.6)	68 (81.0)	0.09
Side effects or patient request	19 (10.2)	13 (12.7)	6 (7.2)	0.23
Others	3 (1.6)	2 (1.9)	1 (1.3)	1.0
Best Response to fulvestrant				
Complete response	2 (1.1)	2 (1.9)	0	0.50
Partial response	24 (12.9)	14 (13.8)	10 (11.9)	0.82
Stable disease	106 (57)	65 (63.7)	41 (48.8)	0.05
Progressive disease	53 (28.6)	21 (20.6)	32 (38.1)	0.009
Unknown	1 (0.5)	0	1 (1.2)	0.45
Received chemotherapy post fulvestrant	112 (60.2)	57 (55.9)	55 (65.5)	0.22

**Table 3 cancers-13-04163-t003:** Time to progression and overall survival of patients treated with fulvestrant.

	TTP		OS	
Variables	Months (95% CI)	*p* Value	Months (95% CI)	*p* Value
Overall	8 (5.6–10.4)		21 (16.0–26.0)	
Primary resistance	7 (2.8–11.2)	0.098	32 (17.2–46.8)	0.592
Secondary resistance	9 (6.1–11.9)		21 (15.6–26.4)	
Chemotherapy before fulvestrant	6 (4.5–7.5)	0.039	21 (10.7–31.3)	0.519
No chemotherapy before fulvestrant	12 (8.7–15.3)		21 (14.8–27.2)	
Chemotherapy after fulvestrant	-		34 (30.4–37.6)	<0.001
No chemotherapy after fulvestrant	-		8 (4.2–11.8)	
50 or younger	10 (4.5–15.5)	0.829	32 (23.3–40.7)	0.500
51 or older	8 (5.9–10.1)		21 (16.3–25.7)	
Visceral Metastasis	7 (5.1–8.9)	0.142	18 (14.1–21.9)	0.029
No Visceral Metastasis	11 (7.9–14.1)		32 (23.0–41.0)	
≤2 previous therapies	12 (9.4–14.6)	0.015	26 (16.0–36.0)	0.067
≥3 previous therapies	6 (5.1–6.9)		16 (10.5–21.5)	

**Table 4 cancers-13-04163-t004:** Cox proportional univariate and multivariate analysis of variables correlated with time to progression.

Variables	Univariate		Multivariate	
	HR (95% CI)	*p* Value	HR (95% CI)	*p* Value
Age < 51	0.952 (0.597–1.517)	0.835		
WHO PS < 2	0.89 (0.53–1.49)	0.680		
Comorbid Illness	0.78 (0.55–1.10)	0.170		
History of early-Stage Breast cancer	0.790 (0. 524–1.191)	0.260		
Secondary Endocrine Resistance	0.728 (0.493–1.077)	0.112	0.717 (0.480–1.071)	0.104
Lack of chemotherapy prior to Fulvestrant	0.710 (0.506–0.996)	0.047	0.802 (0.540–1.191)	0.274
Non-Visceral Metastasis	0.783 (0.557–1.099)	0.158	0.701 (0.495–0.994)	0.046
≤2 previous therapies	0.672 (0.480–0.940)	0.020	0.775 (0.517–1.161)	0.216
Combination therapy	0.714 (0.462–1.104)	0.130	0.745 (0.468–1.187)	0.216
Time to Fulvestrant	0.732 (0.523–1.024)	0.068		

**Table 5 cancers-13-04163-t005:** Cox proportional univariate and multivariate analysis of variables correlated with overall survival.

Variables	Univariate		Multivariate	
	HR (95% CI)	*p* Value	HR (95% CI)	*p* Value
Rural City	0.81 (0.59–1.15)	0.197	0.80 (0.58–1.12)	0.20
Age	0.857 (0.542–1.353)	0.507		
WHO PS < 2	0.538 (0.358–0.806)	0.003	0.73 (0.48–1.11)	0.15
Comorbid Illness	0.869 (0.633–1.193)	0.387		
Secondary Cancer	0.70 (0.42–1.16)	0.172	0.65 (0.38–1.10)	0.11
No Smoking	0.86 (0.62–1.21)	0.402		
History of Early-Stage Breast cancer	0.940 (0.630–1.402)	0.760		
Primary Endocrine Resistance	0.90 (0.60–1.33)	0.597		
Chemotherapy prior to Fulvestrant	0.901 (0.653–1.243)	0.525		
Chemotherapy post fulvestrant	0.425 (0.305–0.593)	<0.001	0.32 (0.23–0.47)	<0.0001
Non-Visceral Metastasis	0.701 (0.506–0.970)	0.032	0.70 (0.50–0.97)	0.03
≤2 previous therapies	0.747 (0.544–1.026)	0.072	0.76 (0.55–1.05)	0.10
Combination therapy	0.67 (0.44–1.05)	0.078		
Clinical Benefit	0.574 (0.402–0.819)	0.002	0.44 (0.30–0.65)	<0.0001

## Data Availability

All data presented in this study is publicly not available and data access will require approval form the University of Saskatchewan Biomedical Ethics Board and Data Access Committee of the Saskatchewan Cancer Agency.

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
