# Peer review of "Efficacy of Fulvestrant in Women with Hormone-Resistant Metastatic Breast Cancer (mBC): A Canadian Province Experienceâ€"

_cancers, 2021, doi:10.3390/cancers13164163_

Round 1
Reviewer 1 Report
GENERAL COMMENTS
The Authors report a real-world study on the efficacy of Fulvestrant in a numerically meaningful cohort of patients with HR+ advanced breast cancer In particular they compared outcome in patients treated in early (≤ 2 ) vs later (≥ 3) lines and observed as expected, that TTP and OS were significantly longer in patients treated in earlier lines . However line of treatment was not an independent determinant in multivariate analysis either for TTP and OS while only visceral involvement was independently associated with TTP and OS and clinical benefit rate and administration of chemotherapy after Fulvestrant were independent predictors of OS.
A number of real-world studies on Fulvestrant monotherapy beyond those cited by the authors, including even larger samples than that of the present study which dealt with the efficacy of Fulvestrant according to the line of treatment has been published, all reporting a better outcome in earlier vs later lines A similar result i.e. the relevance of early treatment in univariate but not in multivariate analysis has been previously reported by Palumbo et al (Ther Adv Med Oncol 2019, Vol. 11: 1– 13). The majority of the studies report visceral disease as the main predictor of Fulvestrant efficacy as shown in the present manuscript.
SPECIFIC COMMENTS
The Authors show that administration of chemotherapy after Fulvestrant is independently associated with OS however this variable does not reflect Fulvestrant efficacy.
Clinical benefit rate has not been reported in the Results but the variable is included in multivariate analysis for OS in table IV. It should be reported also in the Results section.
Authors should specify the outcome of patients treated with Fulvestrant in association with targeted therapies since they represented about 20% of patients treated in ≤ 2 line of therapy
Did patients with HR+/HER2+ tumors receive anti HER2 treatment with Fulvestrant?
Table 1 is too busy. I suggest to split patient and tumor characteristics and response to treatment in 2 separate tables.
Table 2 I suggest to report overall data at the head of the table
Table 3: what do the authors intend for early stage breast cancer ? Do they mean that the patients had recurrent breast cancer?
Author Response
Reviewer 1
General Comments
- A similar result i.e. the relevance of early treatment in univariate but not in multivariate analysis has been previously reported by Palumbo et al (Ther Adv Med Oncol 2019, Vol. 11: 1– 13).
We have added citation to Palumbo et al (Ther Adv Med Oncol 2019, Vol. 11: 1– 13) in the revised manuscript (Reference 19).
Specific comments
- Clinical benefit rate has not been reported in the Results but the variable is included in multivariate analysis for OS in table IV. It should be reported also in the Results section.
Clinical benefit definition has been included in the revised manuscript in the “Definition section” (page 3) and clinical benefit rates have been added in the first paragraph of result section (page 4).
- Authors should specify the outcome of patients treated with Fulvestrant in association with targeted therapies since they represented about 20% of patients treated in ≤ 2 line of therapy
Outcome of patients who were treated with fulvestrant in combination with targeted therapies has been included in 3.1 survival section of the results (page 4).
- Did patients with HR+/HER2+ tumors receive anti HER2 treatment with Fulvestrant?
Patients with HR+/HER2+ tumors received anti HER2 treatment with chemothetapy or endocrine therapy including fulvestrant. It has been specified in the result section of the revised manuscript (page 4).
- Table 1 is too busy. I suggest to split patient and tumor characteristics and response to treatment in 2 separate tables.
Thanks very much for this recommendation. We have divided table 1 into 2 tables. Table 1 provides information on patients’ demographic and tumor characteristics and table 2. provides treatment information.
- Table 2 I suggest to report overall data at the head of the table
We have added the survival data of the entire cohort in the 1st row of the table of table 3 (previously table 2).
- Table 3: what do the authors intend for early stage breast cancer? Do they mean that the patients had recurrent breast cancer?
We have changed the variable name to “history of early stage breast cancer” to clarify that these women previously had early-stage breast cancer and developed recurrent disease.
Reviewer 2 Report
#Abstract:
no changes needed
#Introduction: the role of fulvestrant in further lines (after CDKi) should be pointed out more. Even the role of fulvestrant in combinatons (later lines) and the role of new agents (alpelisib) should be mentioned.
#Methods: no changes needed
#Discussion: no changes needed
#Figures: please add legends to the figures/graphs. short description with numbers would improve readability.
Author Response
- Introduction: the role of fulvestrant in further lines (after CDKi) should be pointed out more. Even the role of fulvestrant in combinatons (later lines) and the role of new agents (alpelisib) should be mentioned.
Thanks very much for the suggestion. We have added information about combination of fulvestrant and CDK 4/6 inhibitors and alpelisib along with a reference in the revised paper (reference no. 14).
- Figures: please add legends to the figures/graphs. short description with numbers would improve readability.
We have added figure legend for figure 2B and 3B and provided information on survival in the legend.
Round 2
Reviewer 1 Report
The Authors have answered to most of the points raised.